# Porcine Reproductive and Respiratory Syndrome Modified Live Virus Vaccine: A “Leaky” Vaccine with Debatable Efficacy and Safety

**DOI:** 10.3390/vaccines9040362

**Published:** 2021-04-09

**Authors:** Lei Zhou, Xinna Ge, Hanchun Yang

**Affiliations:** Key Laboratory of Animal Epidemiology of Ministry of Agriculture and Rural Affairs, College of Veterinary Medicine, China Agricultural University, Beijing 100193, China; Leosj@cau.edu.cn (L.Z.); gexn@cau.edu.cn (X.G.)

**Keywords:** porcine reproductive and respiratory syndrome virus (PRRSV), modified live virus (MLV) vaccine, attenuation, heterologous cross-protection, safety, reversion to virulence, recombination

## Abstract

Porcine reproductive and respiratory syndrome (PRRS) caused by the PRRS virus (PRRSV) is one of the most economically important diseases, that has significantly impacted the global pork industry for over three decades, since it was first recognized in the United States in the late 1980s. Attributed to the PRRSV extensive genetic and antigenic variation and rapid mutability and evolution, nearly worldwide epidemics have been sustained by a set of emerging and re-emerging virus strains. Since the first modified live virus (MLV) vaccine was commercially available, it has been widely used for more than 20 years, for preventing and controlling PRRS. On the one hand, MLV can induce a protective immune response against homologous viruses by lightening the clinical signs of pigs and reducing the virus transmission in the affected herd, as well as helping to cost-effectively increase the production performance on pig farms affected by heterologous viruses. On the other hand, MLV can still replicate in the host, inducing viremia and virus shedding, and it fails to confer sterilizing immunity against PRRSV infection, that may accelerate viral mutation or recombination to adapt the host and to escape from the immune response, raising the risk of reversion to virulence. The unsatisfied heterologous cross-protection and safety issue of MLV are two debatable characterizations, which raise the concerns that whether it is necessary or valuable to use this leaky vaccine to protect the field viruses with a high probability of being heterologous. To provide better insights into the immune protection and safety related to MLV, recent advances and opinions on PRRSV attenuation, protection efficacy, immunosuppression, recombination, and reversion to virulence are reviewed here, hoping to give a more comprehensive recognition on MLV and to motivate scientific inspiration on novel strategies and approaches of developing the next generation of PRRS vaccine.

## 1. Introduction

Porcine reproductive and respiratory syndrome (PRRS), characterized as reproductive failure in breeding pigs and respiratory distress in pigs of all age, is one of the costliest diseases disturbing the global swine industry [1,2]. It was initially reported as a “mystery” disease in the United States in the late 1980s and then outbreaks with similar clinical symptoms were also documented in Western European countries in 1991 [3,4]. Each type of PRRS virus (PRRSV), the causative agent, spread rapidly in its respective continent and eventually widely transmit to the most pig producing countries [2,5]. Subsequently, many virulent strains, quite distinct from early prototype strains, have been continually identified in the United States, China, and Eastern European counties [2,6,7]. Especially in 2006, an unparalleled, large-scale, atypical PRRS outbreak caused by the highly pathogenic variants was documented in China, later in Vietnam, and other Southeast Asian countries [8,9,10]. This event has reformed the concept of pathogenicity and the economic impact of PRRSV. Nowadays, PRRSV remains ongoing through the swine population globally [11].

PRRSV is an enveloped, single-stranded positive-sense RNA (+ssRNA) virus, belonging to the family *Arteriviridae*, genus *Porartevirus* [12,13]. There are two species PRRSV-1 (type 1) and PRRSV-2 (type 2), which only share approximately 60% nucleotide sequence identity, and they are recently classified as *Betaarterivirus suid* 1 and *Betaarterivirus suid* 2, respectively, in the genus *Betaarterivirus* (EC 51, Berlin, Germany) (https://talk.ictvonline.org/taxonomy/p/taxonomy-history?taxnode_id=20171832, accessed on 1 July 2019) [14,15,16]. Based on the phylogenic analysis of the ORF5 gene, PRRSV can be divided into at least nine distinct genetic lineages within type 2 virus, and 3 subtypes within type 1 virus [17,18]. A nearly worldwide epidemic has been sustained by a set of emerging and re-emerging strains, attributed to its high-frequency mutation (reported evolutionary rate of 4.7–9.8 × 10^−2^/site/year) and recombination [19,20,21]. As PRRSV continues to rapidly spread in pig-raising regions worldwide, and its prevalence in the herds remains high, PRRS prevention and control are still the top priorities for pig farms.

Since the first animal vaccination was documented in the year 1872, the vaccine has been widely used for preventing and controlling infectious disease in livestock [22]. As one of the main tools to improve animal health and to reduce/limit pathogens transmission, the vaccine is desired to increase the production of livestock in a cost-effective manner. In addition, vaccinations are also considered to play important roles in reducing antimicrobial use and avoiding the emergence of antimicrobial resistance, as well as improving animal welfares [23]. A modified live virus (MLV) vaccine, the first commercial PRRS vaccine, was launched in the United States in 1994 [24]. Then, the PRRS MLV vaccine has been widely used for almost three decades (Table 1), and it is the major commercial vaccine that can successfully induce a protective immune response against the homologous virus and help in reducing the clinical sign and virus shedding during the heterologous viruses infection. However, it fails to confer sterilizing immunity against various field viruses and cannot provide solid protection against heterologous field strains [1,11,24,25]. Since the PRRS MLV is a leaky vaccine that can prevent the development of disease symptoms, but do not protect against infection and the onwards transmission of pathogens. As well, as in the field virus, MLV can still replicate in a subset of monocyte-derived cells of the host and modulate the immune response, as well, it has the potential issues of reversion to virulence and recombination with field strains, its safety has significantly been concerned [26,27]. Considering that the development and commercialization process of a novel PRRS vaccine cannot always match the speed of mutation and recombination in field strains, the chance for a commercial vaccine to provide homologous protection is limited. Thus, in the field of veterinary practices and pig producers, there are some debates regarding if it is valuable or necessary to use the PRRS MLV, given its leaky characterization [28]. To provide insights on vaccination efforts and the safety of PRRS MLV, the recent advances and opinions on MLV attenuation, protection efficacy, and safety concerns, as well as next-generation vaccine design are reviewed here.

## 2. Attenuation of PRRS MLV

Currently, almost all commercially available MLVs were attenuated by continuous passaging of PRRSV in certain cell lines and produced by the cell culture. Most cell lines used for PRRSV attenuation were derived from the African green monkey kidney cell MA-104, such as MARC-145, CRL11171, and CRL2621a, which are also used for PRRSV isolation and viral loads titration [29,30,31]. Other cells lines, such as PK or BHK cells engineered to express the porcine CD163 molecule, have also been developed for PRRSV attenuation and vaccine production [32]. Although PRRS MLV has been used widely in the field for more than 25 years, the mechanism of attenuation is still unknown. As PRRSV is blindly passaged in cell lines to attenuate, that results in the accumulation of its genomic multiple mutations, it leaves the difficulty of investigating the molecular basis of attenuation, even though the genomic sequences of attenuated viruses can be easily obtained and compared with those of their parental strains [33,34]. By comparing genomic sequences of five pairs of virulent parental/attenuated strains of PRRSV-2, previous analyses have shown that the mutation rates of envelope-associated structural proteins were higher than those of most non-structural proteins, and the E protein had the highest mutation rate, but no mutation site conserved among all the attenuated strains was found [34,35]. Thus, it was considered that the PRRSV attenuation is a multigenic effect [34,35]. Moreover, the reverse genetic operation has also been carried out to swap fragments between the wild-type virus and MLV to explore the virulence-related regions of the PRRSV genome. Kwon et al. used a highly virulent PRRSV infectious clone (FL12) and vaccine strain PrimePac to construct a series of chimeric viruses and identified the NSP3–8 and GP5 proteins as the major virulence determinants, as well NSP1–3, NSP10–12, and GP2 as minor virulence determinants [36,37]. In recent years, bioinformatics-assistant synthetic biotechnology was employed for rapid and highly efficacious attenuation of various RNA viruses, through increasing the number of codon pairs that are underrepresented in the coding sequences of the host, known as codon pair deoptimization, and creating unfavourable conditions for viral protein production, processing or folding [38,39,40,41,42]. Considering that the mutation sites are widely distributed in the genome of attenuated PRRSV, we wonder whether there is the possibility that the codon and codon pair changes in PRRSV genomes during cell passage are attributed to the attenuation. However, the analyses based on the codon adaption index (CAI) [43], relative codon deoptimization index (RCDI) [43], similarity index (SiD) [44], and codon-pair bias (CPB) values [38] were conducted on an attenuated strain of HP-PRRSV JXA1, indicating no significant codon and codon pair usage differences through passages on MARC-145 cells (unpublished data). If the knowledge gap of PRRSV attenuation can be filled, this will not only facilitate the fast attenuation of novel emerging field strains but also provide some clues for new strategies to reduce the risk of reversion to virulence.

## 3. Protection Efficacy of PRRS MLV

### 3.1. Protective Mechanism

PRRSV infects a subset of monocyte-derived cells with CD163 expression, such as primary pulmonary alveolar macrophages (PAMs) and pulmonary intravascular macrophages (PIMs), where the virus replication can cause dysfunction or even cell death by necrosis and apoptosis [45]. Meanwhile, PRRSV markedly suppresses the innate immune response and induces inflammatory injury by a variety of mechanisms. As well, PRRSV can cross the maternal-fetal interface (MFI) in the pregnant sow to infect fetuses, leading to reproductive failures [46]. Attenuated PRRSVs infect pigs and cause milder illnesses compared with the virulent wild-type counterparts from which they are derived. Meanwhile, they amplify the amount of antigen available for inducing an immune response in pigs. Since the replication of PRRS MLV mimics that of wild-type virus, the host immune response resembles what occurs after a viral natural infection. However, this is not observed in either inactivated or subunit vaccines [47].

Generally, vaccine-elicited protection relies on the vaccine-induced memory B and T cells. During the PRRSV infection, memory B cells against viral structural and nonstructural proteins are confirmed to be present before viremia is cleared [48]. Even though memory B cells appear to be quite abundant, it has been regarded that there was no anamnestic response to the viral challenge [49]. In the previous studies, almost all challenges were performed before the initial infection of MLV has been completely resolved. In these cases, the protective effect could be due to an ongoing immunity response from the first exposure, before the sterilizing immunity is established. Recently, Rahe et al. created a PRRSV nsp7-specific B cell tetramer to facilitate the detection of PRRSV-specific memory B cells in the lymphoid tissues through a long-term vaccination-challenge study and found that the PRRSV-specific memory B cell response is long-lived in the blood after vaccination and they can be boosted during a live virus re-exposure [50,51,52]. However, it is still too early to conclude whether protection is entirely dependent on PRRSV-specific memory lymphocytes or not.

For many cases of the virus, neutralizing antibodies (NAs) play a key role in protecting from viral infections [53,54,55]. However, for PRRSV, the protection from antibody-mediated neutralization is not very clear, due to the conflicting data from various studies. In the years just after PRRSV identification, the antibodies against PRRSV were initially thought of as an ineffective component of the PRRSV-protective immune response or even deleterious due to the antibody-dependent enhancement (ADE) concerns [56,57]. Generally, the PRRSV NAs in primary infection mainly appear at 28 or 35 day post-infection (dpi), when viremia has been already resolved. Several monoclonal antibodies (mAbs) targeting antigenic regions corresponding to the putative “major NA epitope” have been found to possess less activity [58]. Moreover, it has been demonstrated that M-GP5 ectodomain-specific antibodies purified from the PRRSV-neutralizing serum could bind to the virus but had no neutralization capability, suggesting that the antibodies binding to ectodomain alone are not sufficient to ensure a complete neutralization of PRRSV [59]. In addition, GP3 but not GP5 and M, is regarded as the major target of NA from sera of PRRSV-1 Lelystad virus infected-pigs [60]. This might be reasonable, as GP2, GP3, and GP4 have been confirmed to form a multi-protein complex that binds to the key receptor CD163, playing an important role in PRRSV infectivity [61,62,63]. In addition, once PRRSV ends its first-round infection into the cells, it can utilize nanotubes and exosomes for intercellular spread, resisting to antibody neutralization [64,65]. One more concern is about the NA testing process. Usually, NA titers of most sera are tested on MARC-145 cells, which is not the real host target cell. Our recent study has found that sera with a high NA titer even around 1:96, tested on MARC-145 cells, cannot completely block the PRRSV infectivity in any dilution when parallelly tested on PAMs [66]. This further indicates that the serum with NA titer tested on MARC-145 cells, might not be guaranteed to have the same level of neutralization capability in vivo. In contrast, some other studies have reported that the passive transfer of homologous neutralizing antibodies was shown to prevent reproductive failure and viral transmission to neonatal piglets [67]. In addition, NA titer has been considered as the best predictor of level and duration of viremia [68]. Meanwhile, high titers of broadly neutralizing activity in naturally infected pigs can provide cross-protection against heterologous PRRSV, which correlated with the clearance of the virus from the circulation and tissues [69,70,71]. Thus, the NAs might help reduce PRRSV infection, but cannot completely block the infection, implying that PRRSV might use some strategies to antagonize and escape it. Future studies are still needed to explore the possible mechanism.

Cell-mediated immunity (CMI) and innate immunity are believed to be the major protective mechanism against PRRSV infection [72,73,74,75,76,77]. However, cellular immune response in PRRSV-infected pigs is still poorly understood due to the difficulty of expanding antigen-specific T-cell populations in vitro and the deficiency of tools and reagents to examine antigen-specific responses in vitro or in vivo. Most evaluation on the CMI response was only based on interferon-γ (IFN-γ) ELISPOT or qPCR to test the level of IFN-γ after using PRRSV to stimulate the PBMCs, which were collected from the vaccinated pigs. However, the significance is uncertain, as the specific source of IFN-γ is difficult to further identify. The innate immune response to a PRRSV-1 vaccine (Porcilis^®^ PRRS, MSD) in the first 72 h post-vaccination (hpv) has been investigated through comparing of the PBMC transcriptome profiles at 6, 24, and 72 hpv, between vaccinated and unvaccinated pigs. The results showed that the MAP kinase activity, TRIF-dependent toll-like receptor signaling pathway, T-cell differentiation, and apoptosis were positively regulated, meanwhile, JAK-STAT pathway and regulation, TRAF6-mediated induction of NF-kB and MAPK, the NLRP3 inflammasome, endocytosis, and interferon signaling were downregulated during the early stage of PRRSV vaccination [78,79]. However, the detailed “cross-talk” among infected macrophages and B/T cells, related to the activation of the humoral and cellular immunity, has not been well investigated yet. Thus, the basic research on the mechanism of immune protection is the prerequisite for further improving the cross-protection efficacy of the PRRS MLV vaccine, especially against the heterologous strains.

Pigs are susceptible to PRRSV by direct or indirect exposure, via intranasal, oral, intrauterine, vaginal, and intramuscular routes [80,81]. Once an outbreak occurs, PRRSV tends to circulate within pig herds indefinitely. PRRSV-persistent pigs and the continually induced-susceptible animals can drive endemicity. Vaccination is usually considered as a relatively safer route to reach “herd immunity” [82]. To evaluate vaccine-afforded protection, vaccine trials routinely assess the efficacy at the individual level through immunological, virological, and pathological parameters. For an individual pig, the main objective of vaccination is to protect the animal from the infection and to lessen the clinical symptoms, thereby improving the health level of vaccinated pigs. While at the population level, the efficacy cannot be only evaluated in virological terms [83]. In epidemiological terms, the goal of vaccination is to decrease or even stop viral transmission within a swine farm and reduce infection-related economic losses [80,84,85]. On one side, the PRRS MLV vaccination can reduce the susceptibility of injected pigs, and on the other side, it also decreases the contagiousness of the individuals, by shortening the shedding period and reducing the viral load. Even the heterologous vaccination can decrease the duration of viremia and viral load, resulting in the reduced viral shedding. Several studies have assessed the R0 value of PRRSV transmission in the vaccinated and naïve pigs in the vaccination-challenge trials. For example, in two early studies, PRRSV-1 MLV vaccination could significantly reduce the R0 value (2.78 to 0.53 and 5.42 to 0.30, respectively), when inoculated with PRRSV-1 field strains with 93.4% or 92.7% of nucleotide similarity with the MLV, respectively [84,86]. In another study, an estimate of R0 for the vaccinated contact group was approximately 5.0, one half of that observed for the unvaccinated contact group (mode R0 = 10) [87]. Given the pig ages, MLV and inoculated virus were diverse, different models resulted in different R0 values. However, these results consistently suggest that even when a leaky vaccine cannot completely prevent pigs from heterologous infection, it can still have beneficial impacts on the transmission dynamics, contributing to the reduced R0 value in a pig herd.

### 3.2. Homologous Protection

Presently, it is a relative consensus that PRRS MLV has the highest protective efficacy against the genetically homologous virus, compared with commercially available KV or other kinds of vaccines under development. Numerous publications have described the protective efficacy of MLV under experimental or field condition, around the United States, Europe, and Asia [88,89,90,91]. At the individual level, the efficacy of PRRS MLV is generally described as protective against both reproductive failure and respiratory disorders, providing multiple benefits including but not limited to the reduction of clinical signs, lessened macroscopic and microscopic lung lesion and viremia, shortened viral shedding period and reduced secondary bacterial infection [30,92,93,94]. Reports in a gilts vaccination and late pregnant term challenge study have shown that MLV vaccination can improve the reproductive performance in sows and piglet health and overall viability, compared with unvaccinated sows [95,96,97].

Indeed, there is no fixed “cutoff value” for genetic similarity to classify the “homologous” or “heterologous” strains, the efficacy of homologous protection conferred by PRRS MLV is difficult to be predicted by sequence comparison, especially only based on the sequence of GP5. For example, the Chinese HP-PRRSV-derived MLV such as JXA1-R (P80), HuN4-F112, and TJM-F92 were all described to protect piglets from the lethal challenge of homologous strains, showing no obvious body temperature increase nor other clinical signs throughout the experiment [98,99,100,101,102]. This high protection efficacy could also be observed in the field during the initial years of the HP-PRRSV pandemic. In contrast, another study has reported that Lelystad-like MLV only provides partial protection against the field isolate of the same cluster, suggesting that the degree of genetic homology of ORF5 between the MLV vaccine and challenge isolate is not a good predictor for vaccine efficacy [103]. This is reasonable, as there is no evidence to support that the GP5 encoded by ORF5 is the only protection-related viral protein.

### 3.3. Heterologous Cross-Protection

PRRSV is well characterized by its mutability, which continually leads to the generation of novel variants, frequently causing an outbreak or re-outbreak in PRRS-stable herds, in which pigs have been previously vaccinated or acclimatized [1,24,104]. Lack of providing satisfied heterologous cross-protection against the rapidly evolving virus is the obvious deficiency for most PRRS vaccines, not only for MLV. At the individual level, MLV vaccination usually cannot induce sterilizing immunity to completely block the infection of heterologous strains [75,104,105,106,107]. However, many experimental vaccination-challenge trials or field studies have indicated that PRRSV vaccination can provide partial protection against heterologous strains, shown as delaying the onset of viremia, reducing the duration of viral shedding and significantly decreasing viral load throughout infection, not showing severe clinical signs as unvaccinated animals [75,97,106,108,109,110,111,112,113,114].

To investigate the cross-protection efficacy of commercially available PRRSV-1 and PRRSV-2 MLV against each type of virus, serial vaccination-challenge studies in growing pigs and pregnant gilts were carried out by Chae’s group. The clinical signs including body temperature, respiratory scores, viremia, viral shedding, macroscopic and microscopic lung lesion scores, PRRSV-antigen distribution in interstitial pneumonia, and productive performance such as duration of pregnancy, the ratio of stillborn, and numbers of weaning pigs, together with PRRSV-specific IFN-γ secreting cells in PBMC were all evaluated and compared among two types of MLV-vaccinated groups and unvaccinated groups. Generally, their results indicated that PRRSV-2 MLV was capable of providing partial heterologous cross-protection against the PRRSV-1 virus, but PRRSV-1 MLV was ineffective against PRRSV-2. Importantly, they also found that either PRRSV-1 or PRRSV-2 -specific IFN-γ-secreting cells in the PRRSV-2 MLV-vaccinated group were higher than the PRRSV-1 MLV-vaccinated group, which was regarded to attribute to unidirectional cross-protection between PRRSV-1 and PRRSV-2 [101,110,111,112].

The internal-type cross-protection was also widely investigated. Lager et al. have tested the efficacy of Inglevac PRRS^®^ MLV (from Boehringer Ingelheim, Ingelheim am Rhein, Germany) against Chinese and Vietnamese HP-PRRSV heterologous challenge in pigs, to demonstrate if this commercially available MLV in the United States could be used as an aid in the control of HP-PRRSV outbreaks. Their results indicated that vaccination decreased the duration of viremia and viral load, and shortened the time of high fever and reduced macroscopic lung lesions, compared with those of unvaccinated animals [101]. Similarly, after the United States-originated NADC30-like virus was identified to begin an epidemic in China, the cross-protection efficacy of commercially available vaccines against NADC30-like field strains was investigated by several research groups [107,115,116,117,118]. In our study, two commercial vaccines (JXA1-R and Inglevac PRRS^®^ MLV) and an attenuated low pathogenic strain HB-1/3.9-P40 were used to vaccinate pigs with the same dose as 2 × 10^5^ TCID_50_, the data showed that vaccination in all three groups could not fully reduce the severe level of clinical signs and lung lesions caused by the NADC30-like virus. However, the Ingelvac PRRS^®^ MLV appeared to exert some beneficial effects on shortening the period of clinical fever and improving the growth performance of the challenged pigs [107]. The results of partial or limited cross-protection against the NADC30-like virus were also reported by other groups. The limited efficacy of cross-protection from commercial MLV vaccines against NADC30-like viruses might be an important reason that these viruses widely spread and became the predominant PRRSV strains in China [117,119,120,121]. Furthermore, the Fostera^®^ PRRS MLV from lineage 8 of PRRSV-2 is also confirmed to confer partial cross-protection against the heterologous challenge of a virulent PRRSV strain from lineage 3 [122]. To improve the heterologous protection efficacy, some immune boosters or regulators, such as quercetin and Quil A, which are regarded to be able to upgrade the mRNA expression of interferon and many other helpful cytokines, were orally taken or injected together with PRRS MLV. However, any significant improvement in heterologous cross-protection was not observed [123,124]. Some typical vaccination-challenge (homologous or/and heterologous) studies on different types of MLV are summarized in Table 2.

Given the extensive genetic and antigenic variation of PRRSV, most situations in the field can be considered as a “heterologous challenge”, as the field strains are more or less different from the commercial vaccine strains. Thus, improving the heterologous or even providing broadened cross-protection is one of the major requirements for designing a perfect PRRS vaccine. However, the unclearness of the mechanism on immunological protection greatly hinders the progress of PRRS vaccine development.

## 4. Concerns about MLV Safety

Since all PRRS MLVs replicate in the host, there are various concerns on their safety, including transmission, immunosuppression, reversion to virulence, and recombination.

### 4.1. Transmission

It has been demonstrated that pigs can develop viremia for up to 4 weeks following MLV vaccination, leading to the spread of the vaccine virus to naïve animals [133,134]. In our previous experiment on evaluating the protection efficacy of MLV against the NADC30-like virus, viremia of all three vaccinated groups were still detectable at 28 dpv (0 dpi), with the highest mean titer to 10^−4^ TCID_50_/mL in the group of JXA1-R, an attenuated strain derived from HP-PRRSV [107]. In another study, the semen from PRRS MLV-vaccinated boar was also confirmed to shed the MLV virus for 39 days, increasing the risk of spreading the virus through artificial insemination [135]. The transmission of MLV between the virus carrier and susceptible animals might elevate the selection pressure to screen out new variants with more adaptability to the host, resulting in a reversion to virulence and endemic of MLV-like virus in the herd.

### 4.2. Immunosuppression

Similar to their parental strains, MLVs can also replicate in PAMs, which might more or less impair their function as antigen-presenting cells (APCs). As mentioned above, MLV can suppress the interferon pathway in the first 72-h post-vaccination. As well, MLV can upregulate IL-10 in the double-dose vaccinated host, to impact the immune response against CSFV when both vaccines were immunized at the same time or with only a one-week interval [136]. Recently, an IFN-inducing PRRSV isolate A2MC2, which is genetically close to VR-2332 and Ingelvac PRRS^®^ MLV with 99.8% identity on the nucleotide sequence, has been evaluated as an MLV candidate, when it was conducted on MARC-145 cells by up to 90 serial passages [137]. This was demonstrated as a non-shedding MLV that could protect pigs against the challenge with VR-2385 and also reduced the nasal shedding of pigs infected with the highly virulent strain MN184. Similarly, an artificially designed and synthesized virus PRRSV-Con could also induce type-I IFNs in cell culture [138]. These data raise a hope to design a novel MLV with the ability to elicit innate immunity during immunization, leading to reduced viremia and viral shedding.

### 4.3. Reversion to Virulence

PRRS MLV has the potential of reversion to virulence. The inability of the current PRRS MLV vaccines on conferring sterilizing immunity against field strains may promote viral mutation to adapt to the host environments and escape from an immune response. In the last 25 years, since the first PRRS MLV was licensed to the market, the MLV vaccines have been used widely both in the United States and China, the field isolates with identical nucleotide sequences to the vaccine viruses were frequently discovered, especially after HP-PRRSV-derived MLVs were overused in China [139,140]. Jiang et al. isolated three field strains, sharing the highest nucleotide similarity and identified nucleotide mutation sites with HP-PRRSV-derived vaccine strain JXA1-R and being able to cause high fever and mortality in the inoculated pigs, indicating the reversion to fatal virulence [141]. The high reversion of an HP-PRRSV-derived vaccine candidate was further confirmed in one of our co-operated projects. Two HP-PRRSV strain JX143-derived vaccine candidates at passage 87 (JXM87) and 105 (JXM105) were assessed for reversion to virulence through a reverse passage test in pigs. At the test of the third passage, the increased clinical signs and lung lesions were observed in the pigs inoculated with both candidates, and the elevated viremia was accompanied by the increased clinical severity, indicating that both candidates regained virulence [142]. In most reverse passage trials, pigs were usually intramuscularly inoculated to ensure an accurate dosage of inoculum and to facilitate the experiment. However, intranasal inoculation might be a better way to mimic the real transmission in herds, most importantly, it might increase the selection pressure for the vaccine virus to adapt in vivo. This hypothesis has been confirmed in our recent study, that an attenuated vaccine candidate strain JXwn06-P80, which was derived from HP-PRRSV JXwn06, was serially passaged in piglets through intranasal inoculation to mimic its infection, adaption, and evolution in the host and the virus regains its fatal virulence at the 9th passage (unpublished data). These studies suggest that more strict reverse passage trials should be considered, when evaluating the safety of MLV, especially for the HP-PRRSV-derived MLV. The molecular basis of reversion to virulence for PRRSV remains unclear. Even though the sequencing and reverse genetic operation have been employed to figure out the factors related to virulence, the majority of identified mutation sites might only contribute to the virulence change in a specific strain, undoubtedly this increases the difficulty of designing some strategies to reduce or avoid the reversion of MLV to virulence.

Compared with the DNA virus, the RNA virus shows lower replication fidelity and higher mutability [143]. In the early study of poliovirus and foot and mouth disease virus (FMDV), the Ribavirin-screened viruses present the increased fidelity, which benefit the stability of the viral genome, additionally, reducing the diversity of quasispecies, resulting in viral attenuation as well [144,145]. To overcome the risk of virulence reversion of MLV, a Ribavirin-selected high fidelity vaccine candidate derived from VR-2332 was developed, and compared with the parental strain, showing the increased genetic and phenotypic stability even after three sequential passages in pigs [146]. Although this is a good optimization for MLV, the commercial use of this method is still on the way.

### 4.4. Recombination

An early study led to a proposal that nidoviruses replicate in a discontinuous and nonprocessive manner, resulting in free RNA intermediates, which could be used in RNA recombination via a copy-choice mechanism [147,148]. This occurs in PRRSV when two or more strains co-infect in cell culture or animals. The recombination of PRRSV was first confirmed in the laboratory using two different PRRSV strains to co-infect MA-104 cells [149]. Since then, numerous field isolates with the characterization of recombination among field strains and MLVs have been recognized [150,151,152,153,154]. Based on our continual monitoring on field isolates and recombination analyses on all available whole genomic sequence of Chinese PRRSV-2 in the GenBank database, we found that during the year 2012 to 2017, the ratio of recombined virus significantly increased year by year, which is consistent with the results currently published by Jiang et al. [121]. Interestingly, in our laboratory, four different recombinant field strains were continuedly isolated from a single pig farm (a large multi-site production system). Among these isolates, two strains HeN1401 and HeN1601 were identified as recombinant from NADC30-like virus and an MLV-like strain evolved from HP-PRRS MLV TJM-F92, one strain HeN1501 belonged to a recombinant virus from two MLV-like viruses evolved from HuN4-F112, and importantly the HeN1301 was a recombinant virus from two MLVs (TJM-F92 and HuN4-F112), which were successively used on this farm [155,156,157].

Similarly, in France, a recombinant virus was isolated from a farm, where two PRRSV-1 MLVs were successively vaccinated with a few weeks apart. This field recombinant strain, derived from PRRSV-1 MLV Unistrain^®^ PRRS as the major backbone with three recombined fragments from another vaccine strain Porcilis^®^ PRRS, has been identified to show the increased viremia and transmission capability, compared with the two parental strains [157]. Later in July 2019, another PRRSV-1 recombinant virus from two MLVs (Amervac and 96V198) was reported to cause an outbreak in a Danish boar station and then further spread to 38 herds through the provided semen. Meanwhile, this recombinant virus shares a high genetic identity with its two MLV parental strains, and shows highly transmissible characterization and the increased pathogenicity [158]. These results warn us that different MLVs should not be used on the same farm and it should be carefully planned if pig producers change the MLV strain applied on the farm, to reduce the risk of recombination. Given the low genomic similarity between PRRSV-1 and PRRSV-2, the inter-species recombination has not been reported yet.

As recombination events frequently occur among field strains and MLVs, recombination has been regarded as an important force to fasten the PRRSV evolution. Although the breakpoints and recombinant patterns, a piece of valuable information for understanding viral evolution, can be easily figured out through sequencing, their contribution to virulence and antigenic characterization of recombinant remains hard to know. Based on a previous study on mouse hepatitis virus (MHV), a member of nidoviruses, the RNA recombination was found to be random, if there is no selection pressure, but only certain recombinants with higher adaption capability in the cell culture can be finally selected out, showing “hotpots” in the genome for recombination [159]. Similarly, for PRRSV, the recombinants selected out should be the “winner” of competition on adapting the host environment, escaping from the immune response or transmission in the herd. One previous pathogenic evaluation study in our laboratory can partially support this hypothesis. The field isolate TJnh1501, which was identified as a recombinant between NADC30-like virus and MLV-like derived from vaccine strain TJM-F112, exhibited the increased pathogenicity in comparison to its parental viruses, even though it did not reach the level of wild-type HP-PRRSV [156]. Obviously, recombination can drive the evolution of PRRSV, raising the risk of reversion to virulence, thus, reducing the chance of recombination should be one direction to improve the safety of the PRRS vaccine.

## 5. Genetically Modified Live Virus Vaccine

Since the first RNA-launched infectious cDNA clone of the Lelystad virus, the PRRSV-1 prototype, was successfully constructed, more than 20 distinct PRRSV infectious clones have been generated [24]. With the platform of reverse genetic operation, it is possible to artificially edit the virus by point mutations, truncations, gene insertion or fragment swapping between different strains, to identify the virulence factors, cross-protection antigens, and other factors related to protective efficacy and vaccine safety, which will contribute to the development of PRRS vaccine. An informative table summarizing the knockouts and knockdowns of viral genes and their influence on the viruses was presented in a previous review [24]. Furthermore, various strategies have been documented based on the reverse genetic operation. In order to improve the heterologous protective efficacy, viral genes or clusters of genes from strains with different antigenic characterizations were swapped to create the chimeric viruses or the recombinant viruses carrying DNA shuffled fragments or the conserved fragment of sequence from multiple strains were constructed. In addition, codon pairs de-optimization was also used for rapid attenuation of the virus. As well, foreign fragment including B-cell epitope, protective antigen, and adjuvant cytokines were inserted into the genome of PRRSV to create a marker vaccine, multivalent vaccine or protective efficacy-improved vaccine. These novel strategies and approaches to develop the next generation of vaccines have been well-reviewed before [11,57].

## 6. Principles of MLV Utilization on a Swine Farm

Even though no perfect PRRS vaccine is currently available, MLV is still a useful tool in the “PRRS elimination toolbox”. Some items should be listed in its “manual” to optimize the MLV utilization and to place the safety issue. The first one which should be included, is never putting all our faith in the vaccine, as MLV does not guarantee the success of PRRS control. Biosecurity, serological and genetical evolutionary monitoring, secondary bacterial infection control, and production management on pig herds are also essential parts of the strategy. The vaccination program is usually tailored on different farms based on the PRRS epidemic situation, facility, and management level of the farm. However, the common principles for PRRS MLV utilization should be followed: (i) to keep PRRSV-free in breeding herds, especially in boars with no vaccination; (ii) for the PRRS-unstable herds, MLV can be used based on the infectious situation of PRRSV, but multiple strains of MLV should not be applied on one farm to reduce the risk of recombination; (iii) efforts should be made to establish PRRSV-negative gilts; (iv) overuse of PRRS MLV should be avoided and the MLV with lower viremia and shorter shedding period should be chosen, leaving enough time for vaccinated gilts to “cool down” before they are introduced into sow herds, to reduce the risk of recombination and reversion to virulence. If there is a sufficient consideration and operation to control the risk, even the leaky MLV vaccines can practically provide its benefit in reducing clinal signs at the individual level and PRRSV transmission at the herd level.

## 7. Other Kinds of PRRS Vaccine

Another kind of commercially available PRRS vaccine is the killed vaccine (KV). Compared with MLV, KV is safer, as it does not replicate in the vaccinated animals. Protective effects induced by KV are mainly mediated by humoral immune responses. Many commercial and experimental PRRSV KV have been evaluated, and the induced protective immunity may not be satisfactory [47]. In recent years, multiple novel strategies and techniques, such as the use of reverse genetics system, novel adjuvant or delivery way, diverse protein-expressing system, DNA vaccine platform, viral vector expression system, and T-cell epitope vaccine, are widely involved to design a novel PRRS vaccine to solve the “critical issues” of current commercial vaccines, such as variable heterologous cross-protection, reversion to virulence, recombination, and immunosuppression [160,161,162,163,164,165]. Many novel discoveries have been made with the potential to point out a new direction for PRRSV vaccinology, even though no new vaccine candidates are commercially available.

Given the fact that the cross-protection mechanism is still unclear, especially the targets mediating protective immunity and the role of neutralizing antibodies in preventing PRRSV infection are still controversial, the primary hurdle of developing a PRRSV subunit or live vector vaccine is not about how to deliver the viral antigens into the host and increase the immune response, but about which viral antigens should be enrolled.

## 8. Conclusions and Perspectives

In the past two decades, accumulated field experiences and experimental data have indicated that MLV derived from a specific PRRSV strain can provide good protection against the homologous viruses, and partial protection against heterologous strains, which make MLV an adjunct to elimination strategies. However, due to the high mutability and genetic diversity of PRRSV, outbreaks still occur in herds with regular vaccination, which drives the heterologous cross-protection efficacy and safety to two top concerns of PRRS MLV. They are also two key directions to solve for vaccine development in the future. Indeed, the standards for next generation of PRRS vaccine have been discussed and formally proposed on a colloquium for developing the PRRS vaccine, held at the University of Illinois in 2007, which includes: (a) rapid induction of immunity; (b) protection against most currently prevalent PRRSV strains; (c) no adverse outcomes to swine health; and (d) ability to differentiate vaccinated pigs from infected animals [11]. 

To rapidly induce immunity, novel adjuvant and antigen delivery system, vaccination route modification, removal of immunosuppression factors, expressing self-carried cytokine or other immunocompetent factors, and many other strategies have been tried or are being developed. Based on the technological improvement in reverse genetic manipulation, the developing routes for the PRRS DIVA vaccine are clear and practical. Given the mechanisms of PRRSV fast mutation, the recombination and immune response for cross-protection are still far from being fully elucidated, especially the targets of inducing protective immunity and the role of cell-mediated immunity and neutralizing antibodies in PRRSV clearance are still debatable, the development of a novel vaccine with broadened cross-protection and improved safety greatly depends on the future progress of the basic research.

## Figures and Tables

**Table 1 vaccines-09-00362-t001:** Commercially available porcine reproductive and respiratory syndrome (PRRS) modified live virus (MLV) vaccines.

Vaccine	Parental Strain	Species/Type	Lineage	Producer/Developer
Ingelvac PRRSFLEX^®^ EU	94881	PRRSV-1	lineage 1	Boehringer Ingelheim
ReproCyc^®^ PRRS EU	94881	PRRSV-1	lineage 1	Boehringer Ingelheim
Pyrsvac-183^®^	All-183	PRRSV-1	-	Syva
Unistrain^®^ PRRS	VP-046 BIS	PRRSV-1	lineage 1	Hipra
Amervac^®^ PRRS	VP-046	PRRSV-1	lineage 1	Hipra
Porcilis^®^ PRRS	DV	PRRSV-1	lineage 1	MSD Animal Health
Suvaxyn^®^ PRRS MLV	96V198	PRRSV-1	lineage 1	Zoetis
Prevacent^®^ PRRS	RFLP 184	PRRSV-2	lineage 1	Elanco
Ingelvac PRRS^®^ MLV	VR-2332	PRRSV-2	lineage 5	Boehringer Ingelheim
R98	R98	PRRSV-2	lineage 5	Nanjing Agricultural University
PRIME PAC^®^ PRRS+	Neb-1	PRRSV-2	lineage 7	MSD Animal Health
Ingelvac PRRS^®^ ATP	JA-142	PRRSV-2	lineage 8	Boehringer Ingelheim
JXA1-R	JXA1	PRRSV-2	lineage 8	Chinese Center for Animal Disease Control and Prevention
GDr180	GD	PRRSV-2	lineage 8	China Institute of Veterinary Drug Control
CH-1R	CH-1a	PRRSV-2	lineage 8	Harbin Veterinary Research Institute, CAAS
HuN4-F112	HuN4	PRRSV-2	lineage 8	Harbin Veterinary Research Institute, CAAS
TJM-F92	TJ	PRRSV-2	lineage 8	Institute of Special Animal and Plant Sciences, CAAS
Fostera^®^ PRRS	P129	PRRSV-2	lineage 8	Zoetis
PRRSV-PC	PC *	PRRSV-2	lineage 8	China National Pharmaceutical Group

Note: * A chimeric virus between the classical malicious PTK strain of PRRSV and HP-PRRSV strain, constructed by reverse genetic operation.

**Table 2 vaccines-09-00362-t002:** Studies on evaluating cross-protection efficacy of commercial MLV vaccines

MLV	Challenge Virus	Species/Types (MLV/Challenge)	Homologous/Heterologous	Tested Animals	Parameters for Immune Response	Results and Reference
Porcilis^®^ PRRS	PR40/2014	PRRSV-1/PRRSV-1	Heterologous	Piglet	Ab and NAb	Triggered adaptive immunity against highly pathogenic strain, and reduced clinical indicators [125]
Amervac^®^ PRRS	KKU-PP2013	PRRSV-1/PRRSV-2	Heterologous	Piglet	Ab	A certain degree of protection against the PRRSV-2 challenge [126]
Amervac^®^ PRRS	01NP1	PRRSV-1/PRRSV-2	Heterologous	Piglet	Ab/IFN-α, IFN-β and IFN-γ	Upregulated IFN-α, IFN-β, and inflammatory cytokines and reduced PRRSV-2 viremia and number of viremic pigs [124]
Fostera^®^ PRRS	SNUVR090485	PRRSV-2/PRRSV-1	Heterologous	Piglet	Ab/IFN-γ secreting cells	Partial protection from the challenge of heterologous type 1 PRRSV and reduced viremia [111]
HuN4-F112	HuN4-F5	PRRSV-2/PRRSV-2	Homologous	Piglet	Ab and NAb	Protection from the lethal challenge [99]
Ingelvac PRRS^®^ MLV	VR-2332-P6, rJXwn06-P3, rSRV07-P3	PRRSV-2/PRRSV-2	Homologous/heterologous	Piglet	Ab	Partial protection against the homologous and heterologous PRRSV challenge [101]
JXA1-R	HV-PRRSV, NADC-20	PRRSV-2/PRRSV-2	Homologous/heterologous	Piglet	Ab and NAb/IFN-α and IFN-β	Protection from the challenge of HP-PRRSV or NADC-20, induced broadly neutralizing antibodies and enhanced pulmonary IFN-α/β production [90]
Ingelvac PRRS^®^ MLV	10186-614	PRRSV-2/PRRSV-2	Heterologous	Piglet	Ab	No prevention in viral shedding, reduced viral replication, and disease severity [127]
Ingelvac PRRS^®^ MLV/JXA1-R/(HB-1/3.9-P40)	CHsx1401(NADC30-like virus)	PRRSV-2/PRRSV-2	Heterologous	Piglet	Ab	Reduced clinical signs and lung lesions, shortening the period of clinical fever and improving the growth performance (Ingelvac PRRS^®^ MLV) [107]
PrimePac^®^ PRRS	dss	PRRSV-2/PRRSV-2	Heterologous	Piglet	Ab/Treg, IL-10, and IFN-γ	Partial protection against the Thai HP-PRRSV, based on body temperature, levels of viremia, and lung lesion [128]
Ingelvac PRRS^®^ MLV	1-4-4	PRRSV-2/PRRSV-2	Heterologous	Piglet	Ab and NAb/IFN-γ secreting cells (total lymphocytes, NK, CD4^+^, CD8^+^, and γδT cells)	No improvement in the efficiency of cross-protection (adjuvant M. vaccae WCL or CpG ODN), induced virus-specific T cell response (IM vaccination) [129]
Fostera^®^ PRRS	SNUVR090485	PRRSV-2/PRRSV-1	Heterologous	Gilt	Ab/IFN-γ secreting cells	Cross-protection against the PRRSV-1 challenge in late-term pregnant gilts, improved reproductive performance, and induced immunity lasting for 19 weeks at least [130]
Unistrain^®^ PRRS	SNUVR090485, SNUVR090851	PRRSV-1/(PRRSV-1 or PRRSV-2)	Heterologous	Gilt	Ab/IFN-γ secreting cells	Vaccinated pregnant sows with the PRRSV-1 MLV against PRRSV-1, but limited to PRRSV-2 in late-term pregnant gilts [95]
Ingelvac PRRS^®^ MLV	SNUVR090485, SNUVR100059	PRRSV-2/(PRRSV-1 or PRRSV-2)	Heterologous	Sow	Ab/IFN-γ secreting cells	Vaccinated pregnant sows with the PRRSV-2 MLV against PRRSV-2, but not to PRRSV-1 [131]
Unistrain^®^ PRRS/Fostera^®^ PRRS	SNUVR090485, SNUVR090851	(PRRSV-1 or PRRSV-2)/(PRRSV-1 + PRRSV-2)	Heterologous	Gilt	Ab/IFN-γ secreting cells	PRRSV-2 MLV vaccine is more efficacious than PRRSV-1 MLV against the dual heterologous challenge in gilts [132]

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
