# Peer review of "Porcine Reproductive and Respiratory Syndrome Modified Live Virus Vaccine: A “Leaky” Vaccine with Debatable Efficacy and Safety"

_vaccines, 2021, doi:10.3390/vaccines9040362_

Round 1
Reviewer 1 Report
This is a comprehensive review of PRRSV MLV. In this review, the authors summarized the current status and future direction of PRRSV MLV development, which would be very informative to readers. No major critism from this peer reviewer, only two suggestions. One is that the English writing has lots of space to be improved. Another one is that decrease the number of subtitles to make it more compact.
Author Response
Author’s notes to reviewer 1
We really appreciate the reviewer for your time and efforts on reviewing our manuscript, especially for the constructive suggestion on decreasing the number of subtitles. In the revised version we have combined the paragraphs under subtitles 3, 4, 5 and 6 into one subtitle as “3. Protection efficacy of PRRS MLV”. And finally, there are 8 subtitles in the revised version. Meanwhile, we have carefully proofreaded the manuscript several times to correct the grammar errors and misused terms in the manuscript. The modifications have been highlighted in the revised version with track changes.

Reviewer 2 Report
The topic is important and interesting but the manuscript is too unorganised and the numerous language mistakes make the text difficult to read. Some sentences are incomprehensible, a professional language editor or native English speaker should be enrolled to revise the entire text
There is a lot of information in this manuscript and the reader needs to know where to find the different facts. Currently, a lot of facts and views appear under several headings. The text needs a thorough revision with everything sorted in the appropriate section. The repeated discussion about some aspects make the argument less convincing than if it had been put in one section focusing on that particular aspect, and not repeated elsewhere.
For scientific stringency, I advice you to decide on distinct headings that make a logical flow and only include text belonging there under each respective heading. Put the discussion about safety aspects under a separate heading and focus it there.
I would also suggets that you describe relevant aspects (safety and efficacy) for other types of vaccines as well, preferably in one section with a comprehensive discussion about all currently available vaccines, their pro’s and con’s.
Author Response
Author’s notes to Reviewer 2
We deeply appreciate the reviewer’s helpful and constructive comments and suggestions.
- 1. “The topic is important and interesting but the manuscript is too unorganised and the numerous language mistakes make the text difficult to read. Some sentences are incomprehensible, a professional language editor or native English speaker should be enrolled to revise the entire text.”
Response: Thank you for the positive feedback on the importance of this topic as well as the constructive comments. After read the reviewer’s commnents, we also notice there were some concerns on the orginzation of our manuscript. Thus, we first combined the paragraphs under subtitles 3, 4, 5 and 6 into one subtitle as “3. Protection efficacy of PRRS MLV”, and put the “Protective mechanism” forward. The order of subtitles “Principles of MLV utilization on swine farm” and “Other kinds of PRRS vaccine” has been exchanged. Meanwhile the manuscript has been carefully checked several times to correct the grammar errors and unclear terms.
- “There is a lot of information in this manuscript and the reader needs to know where to find the different facts. Currently, a lot of facts and views appear under several headings. The text needs a thorough revision with everything sorted in the appropriate section. The repeated discussion about some aspects make the argument less convincing than if it had been put in one section focusing on that particular aspect, and not repeated elsewhere.”
Response: When we prepared the revision of this manuscript, we noticed this problem. We agree with the reviewer’s comment, some aspects such as the immunosuppression effect of MLV have been descripted or discussed at different sections. We have deleted some sentences with cumbersome repetition at different sections.
- “For scientific stringency, I advice you to decide on distinct headings that make a logical flow and only include text belonging there under each respective heading. Put the discussion about safety aspects under a separate heading and focus it there. “
Response: We agree with reviewer’s suggestion, the heading (subtitles) have been reorganized and the paragraphs with cumbersome repetition have been modified in revised version. The deleted parts can be noticed by “tracking changes”.
- “I would also suggest that you describe relevant aspects (safety and efficacy) for other types of vaccines as well, preferably in one section with a comprehensive discussion about all currently available vaccines, their pro’s and con’s.”
Response: The advantage and disadvantage of killed vaccine have been discussed in Line 496-500. As well, the primary hurdle of developing the subunit or vector vaccine is attributed to the unclearance of protective antigen in PRRSV (Line 508-512).

Reviewer 3 Report
Summary
PRRS is one of the most important diseases in swine industry, while still facing many challenges for virus control, including lack the understanding of pathogenesis and limited vaccine protective efficiency. This review described MLV vaccine issues through comparing immune protection and vaccine safety in different MLV vaccines, which provides comprehensive information on MLV vaccines and future vaccine design strategies.
Specific points
- Table 1, It would be great and complete if adding features for each product.
- Lacking diagram for PRRSV infection and pathogenesis, help readers to understand which viral protein or potential targets are the main vaccine design targets currently or in the future.
- Lacking diagram for MLV vaccines “work-flow” from infection to generate immune protection, it would be helpful to know how does MLV vaccine work and its advantages and potential problems.
- Many spelling and grammatical errors need to be corrected.
Author Response
Author’s notes to Reviewer 3
We would like to thank the reviewer’s positive feedback and meaningful suggestions.
Specific points
- Table 1, It would be great and complete if adding features for each product.
Response: This is really a great suggestion, however there are some difficulty for us to find some addiontional information for each product. As there is no research parallally compared these vaccine products on their protection effciency or safety. Even the sequnce informations of them are limited.Currently we really can not provide any additional data to compare them.
- Lacking diagram for PRRSV infection and pathogenesis, help readers to understand which viral protein or potential targets are the main vaccine design targets currently or in the future.
Response: Follow the reviewer’s suggestion, we have added some brief description and references on the pathogensis of PRRSV in revised version (Line 135-141).
- Lacking diagram for MLV vaccines “work-flow” from infection to generate immune protection, it would be helpful to know how does MLV vaccine work and its advantages and potential problems.
Response: Follow the reviewer’s suggestion, we have discussed the protection mechanisms through the aspect of NA, memory B/T cells,and cell-mediated and innate immunity. However, we still can’t provide the information that which viral protein is solidly related with the protective immune response induced by PRRSV.
- Many spelling and grammatical errors need to be corrected.
Response: Thank you for the comments, we have carefully and thoroughly proofread the manuscript several times to correct the grammar errors and misused terms in the manuscript.

Round 2
Reviewer 2 Report
I find the manuscript vastly improved and much easier to read. However, there are still numerous language errors and I have commented below on some of these as well as some minor things I believe would be useful to clarify. I still recommend a second thorough language proof-reading.
There is a constant mixing/misuse of the definite and the indefinite forms (i.e. the use, or lack of, "the" where appropriate). In addition, some words are erroneously used, such as "curious" on line 123 (I think you mean that you hypothesise this or wonder/question whether this is the case), "premised" on line 146 (premise is not a verb), "researches" on line (research is an uncountable noun), "well characterised" on line 267 (I think you mean that PRRSV is renowned or characterised by its mutability),"considerable" on line 273 (should be replaced by "several" or "many") "potentiality" on line 361 (should be "potential") "interestedly" on line 409 (should be "interestingly") "French" on line 417 (should be "France"), and "marked vaccine" on line 465 (should be "marker vaccine").
I suggest a clear definition of exactly what you mean by "leaky" and not leave this tio the readers' imagination. The same for "unstable herd" (line 479) and "pain point" (line 496, this phrase could be replaced by "critical issues").
A reference should be added for the paragraph on lines 122-131, as it is not clear whether this stems from one or all of the references cited immediately before, or from somewhere else.
On line 151, please clarify if you by "initial infection" mean the infection by the live vaccine strain.
On line 293, please clarify what is meant by "internal-type" cross-protection (how does this differ from the cross-protection discussed in the previous paragraph?)
On line 381, please elaborate (briefly) on how the hypothesis was confirmed in your recent study with unpublished data (as the data cannot be accessed).
On line 477-478, please clarify the first point, do you mean that MLV should be used to keep breeding herds free from PRRSV? How would that be possible, MLV contains PRRSV?
Author Response
Responses to the reviewer
- There is a constant mixing/misuse of the definite and the indefinite forms (i.e. the use, or lack of, "the" where appropriate).
Response: with the help of online program from Grammarly, we have carefully proofread the manuscript to correct the misused definite/ indefinite forms. And the modification can be tracked in the manuscript.
- In addition, some words are erroneously used, such as "curious" on line 123 (I think you mean that you hypothesise this or wonder/question whether this is the case).
Response: "curious" has been replcaced by “wonder”
- "premised" on line 146 (premise is not a verb), "researches" on line (research is an uncountable noun
Response: we agree with it. “premise” has been replaced by “relies on”, “Researches” has been replcaced by “studies”.
- "well characterised" on line 267 (I think you mean that PRRSV is renowned or characterised by its mutability)
Response: Line 288, “characterized of “ has been replaced by “characterized by”.
- "considerable" on line 273 (should be replaced by "several" or "many") "potentiality" on line 361 (should be "potential") "interestedly" on line 409 (should be "interestingly") "French" on line 417 (should be "France"), and "marked vaccine" on line 465 (should be "marker vaccine").
Response: All words mentioned above have been replaced with "many"(Line 294), "potential"(Line 392), "interestingly"(Line 449), "France"(Line 457), and "marker vaccine"(Line 510), respectively.
- I suggest a clear definition of exactly what you mean by "leaky" and not leave this tio the readers' imagination.
Response: “Leaky vaccine” means the vaccine can prevent the development of disease symptoms, but do not protect against infection and the onwards transmission of pathogens. A sentence to set the definition of “ Leaky vaccine” has been added on Line 80.
- The same for "unstable herd" (line 479) and "pain point" (line 496, this phrase could be replaced by "critical issues").
Response: PRRS situation of the herd can be classified as four items, the “unstable” is one PRRS related situation of the herd. “unstable herd” has been replaced as “PRRS-unstable herd” (Line 529) and the “ pain point” has been replaced with "critical issues" (Line 548).
- A reference should be added for the paragraph on lines 122-131, as it is not clear whether this stems from one or all of the references cited immediately before, or from somewhere else.
Response: four reference about the codon and codon pairs have been added (highlighted).
- On line 151, please clarify if you by "initial infection" mean the infection by the live vaccine strain.
Response: Yes, it means "initial infection" of MLV.
- On line 293, please clarify what is meant by "internal-type" cross-protection (how does this differ from the cross-protection discussed in the previous paragraph?)
Response: In previous paragraph (Lse 305-318), it discussed the cross-protection between genetype 1 and 2 viruses.
- On line 381, please elaborate (briefly) on how the hypothesis was confirmed in your recent study with unpublished data (as the data cannot be accessed).
Response: A introduction of our current study on reversion to virulence has been added (Line418-421).
- On line 477-478, please clarify the first point, do you mean that MLV should be used to keep breeding herds free from PRRSV? How would that be possible, MLV contains PRRSV?
Response: It is a misunderstanding. To keep breeding herds free of PRRSV, the PRRS vaccine cannot be used in the herd.
